# *Mastomys* Species as Model Systems for Infectious Diseases

**DOI:** 10.3390/v11020182

**Published:** 2019-02-21

**Authors:** Daniel Hasche, Frank Rösl

**Affiliations:** Division Viral Transformation Mechanisms, Research Program “Infection, Inflammation and Cancer”, German Cancer Research Center, 69120 Heidelberg, Germany; d.hasche@dkfz.de

**Keywords:** *Mastomys coucha*, animal models, papillomavirus, parasites, cancer

## Abstract

Replacements of animal models by advanced in vitro systems in biomedical research, despite exceptions, are currently still not satisfactory in reproducing the whole complexity of pathophysiological mechanisms that finally lead to disease. Therefore, preclinical models are additionally required to reflect analogous in vivo situations as found in humans. Despite proven limitations of both approaches, only a combined experimental arrangement guarantees generalizability of results and their transfer to the clinics. Although the laboratory mouse still stands as a paradigm for many scientific discoveries and breakthroughs, it is mandatory to broaden our view by also using nontraditional animal models. The present review will first reflect the value of experimental systems in life science and subsequently describes the preclinical rodent model *Mastomys coucha* that—although still not well known in the scientific community—has a long history in research of parasites, bacteria, papillomaviruses and cancer. Using *Mastomys*, we could recently show for the first time that cutaneous papillomaviruses—in conjunction with UV as an environmental risk factor—induce squamous cell carcinomas of the skin via a “hit-and-run” mechanism. Moreover, *Mastomys coucha* was also used as a proof-of-principle model for the successful vaccination against non-melanoma skin cancer even under immunosuppressive conditions.

## 1. Introduction

Model systems are indispensable in science, not only to visualize corresponding features of the invisible (Figure 1), but also to provide—in biomedical research—an empirical *bona fide* equivalent for pathophysiological processes of human diseases, such as cancer. Models are only needed as long as we do not yet have full knowledge of what they stand for. When enough certainty is obtained in form of reproducible data sets, the exploited models are obsolete and become substituted by others or by the genuine respective organism for further questions [1]. For instance, Gregor Mendel’s cross-breeding of pea plants represented a model to understand dominant and recessive inheritance, but Thomas Hunt Morgan’s work with *Drosophila melanogaster* became more suitable, since phenotypes could be mapped to a defined region within the chromosomes. Moreover, although the spinning top watched by the two Nobel Laureates Wolfgang Pauli and Niels Bohr (Figure 1) just represents an amusing metaphor of a scientific model for a component of the inanimate matter, namely the electron [2], it nonetheless implicates that some observations in research cannot be represented in any other way than in the form of models.

This becomes even more important when living systems like cells, three-dimensional tumors or even whole organisms with their emergent properties are considered [3,4]. A model also stands as a substitute for an inevitable reductionist approach to comprehending the complexity of an entity (e.g., primary tumors or metastases) on the basis of studying and knowing its parts (e.g., dysregulated signal transduction pathways, driver mutations) [5]. Accordingly, despite current initiatives to replace laboratory animals by sophisticated in vitro systems [6,7], biomedical research without animals as holistic models may fail to fulfill criteria and social demands of translatability of laboratory results into the clinic [8,9]. Conversely, although the “bench to bedside concept”, combined with personalized oncological treatment sounds attractive [10,11], preclinical models are further worthwhile to be funded in order to comprehend fundamental principles of cancer development without current quite obvious and ultimate clinical applications [12,13]. Clonal evolutions within tumors, for instance squamous cell carcinomas, result in tumor heterogeneity which represents an enormous problem for the treatment of cancer patients. Such evolutionary processes starting from initiation to metastasis, as recently shown in the “Confetti” mouse model, can only be obtained in vivo, but not in tissue culture [14].

Certainly, every model that represents a particular in vivo phenotype has its inherent limitations [15,16] and the choice of an animal species can even be decisive for conclusions or consequences to support future research strategies and/or programs [17,18]. A prominent historical example is the treatment of mice with penicillin, to show the therapeutic effect on staphylococcus infections. If hamsters or guinea pigs were utilized during these times, the proof-of-principle would have failed and the launch of antibiotics would have been delayed, since penicillin is highly toxic for both species [19,20]. Especially animal models for studying infectious diseases must be carefully selected. They should have similar routes of infection, should develop analogous symptoms and have to display comparable pathological changes as seen in humans [21].

Researchers introduced a plethora of animal models, such as zebrafish, rabbits, rats, dogs, pigs, goats, cattle and monkeys [22,23,24,25,26]. To be accepted as a valuable preclinical model, however, a scoring system should guarantee their careful selection, by reflecting face validity, complexity and predictability of a disease [27,28]. Nonetheless, the house mouse *Mus musculus* is still the best characterized organism used in biomedical research [29,30]. To create a homogeneous genetic background inbred mouse strains are used [31], a condition that may affect the experimental read-out of these model systems [27,32]. Continuous inbreeding may also result in so-called “spontaneous” models like the immunocompromised nude mouse, routinely used for xenotransplantation for several decades [33].

Nevertheless, research of infectious diseases requires models with a proper immune system to understand the spread and virulence of bacterial and viral infections, as well as the link between innate and the humoral immune responses. This aspect becomes even more important when immunosuppressive treatment is considered, that may lead to reactivation of persistent infections [34,35]. Only animals with a functioning immune system allow the development and testing of vaccination strategies, since prophylaxis is always superior to therapy. While transgenic, knock-in and knock-out mice are also valuable tools to investigate the complexity of physiological processes, natural animal models better reflect the reality with respect to the immunological surveillance of infections [36]. The holistic concept of an organism to understand the impact on the immune system, on cancer development or the efficiency of therapeutic drugs also becomes evident in current microbiome studies [37,38]. The generation of germ-free mice and their subsequent recolonization with defined microorganisms will show the impact of the microbiome or virome on the outcome of diseases [39,40]. Specific research fields and new trends in biomedical research indeed require and develop suitable nontraditional in vivo systems [41,42] as shown recently in the case of the crayfish as a novel model in epigenetics [43] and the naked mole rat as a model for chromosomal stability and senescence [44,45].

In the present overview, we describe the animal model *Mastomys coucha*. Although not yet very common in the scientific community, it has been known for several decades in various research fields and has recently attracted great attention in functional studies of cutaneous papillomaviruses and non-melanoma skin cancer (NMSC) formation.

## 2. Characteristics of *Mastomys* Species

The genus *Mastomys* belongs to the family *Muridae* (subfamily *Murinae*) and is a phylogenetic relative of mouse and rat [46]. It contains eight species referred to as multimammate mice or multimammate rats (also known as African soft-furred rats or African common rats) that can be found all over sub-Saharan Africa [47,48,49]. The animals have brown eyes and are usually covered by a dense agouti coat that is of lighter grey at the belly, although also other color variants, e.g., chamois-colored coat with pink eyes, exist in laboratory *Mastomys* strains (Figure 2A–D) [50]. The strain used at the German Cancer Research Center (DKFZ) for papillomavirus research and in other laboratories for parasitological studies is a variant with a chamois-colored coat and light red eyes, previously known as the GRA-Giessen strain (Figure 2E,F) [50]. With a typical head-and-body length of six to 17 cm and a tail length of six to 15 cm, *Mastomys* weight ranges between 20 and 100 g [51]. In animal housing *M. coucha*—especially the males—can reach weights of more than 160 g.

Like rats the animals do not have a gall bladder, but the females carry the characteristic eight to 18 pairs of mammary glands giving these animals the name multimammate mouse. The different *Mastomys* species can be roughly assigned to certain geographic locations, e.g., Southern Africa for *M. coucha* and *M. natalensis*, whereby the latter is also found in other regions [49]. The distinction of both species by superficial appearance is impossible and needs analysis of molecular markers e.g., by isoelectric focusing of serum proteins [52,53,54] or karyotyping (e.g., *M. natalensis* 2*n* = 32 vs. *M. coucha* 2*n* = 36; Figure 2G) [51,55]. This is why it was originally thought that both are the same species, but actually their mating only produces sterile offspring [56]. Of the several *Mastomys* species, *M. natalensis* and *M. coucha* are the ones that are used the most in biomedical research.

## 3. Housing of *Mastomys* in the Laboratory

The housing of *M. natalensis* and *M. coucha* is comparable to mice (*Mus musculus*) and rats (*Rattus norvegicus*). They develop well under typical animal facility conditions (20–25 °C, 55–65% humidity) and standard diets and have intermediate space requirements. They are very curious and explore wood shavings or paper towels newly introduced in the cage, which they also use for nest-building (Figure 3). Their gestation period is approximately 23 days and the litter size can be in maximum up to 19 [51], although it is usually around 10 newborns with a birth weight of two to three grams (Figure 3A–C). The young are weaned after approximately four weeks with a weight of approximately 12 g, become sexually mature after one to three months and can reach an age of up to 2.5 years [51].

## 4. *Mastomys* as Model Systems in Biomedical Research—Historical Flashback

Moving back in science history and inspecting different research fields, *Mastomys* is definitely an interesting animal species to study many infectious diseases and cancer. As already noticed very early, *Mastomys* is highly susceptible to plague without any resistance to *Yersinia pestis* [57], which 1939 initially led to the breeding of *Mastomys* for plague research at the Medical Ecology Center in Johannesburg, South-Africa. In 1974, a plague outbreak in Zimbabwe (the former Rhodesia) [58] emerged with *Mastomys* as its primary reservoir host which were in turn used to test attenuated *Y. pestis* strains for vaccination [59]. Although previously suggested that there may be different sibling species of *Mastomys* [60], the discrimination between *M. natalensis* and *M. coucha* via chromosomal G-banding, was not successful before 1977 [61] and in 1983 it turned out that the latter is actually sensitive to *Y. pestis* [62,63]. Interestingly, the geographical distribution of *M. coucha* also correlates with human plague, while *M. natalensis* is resistant and predominates in areas without recorded human plague [62,64].

In 1969, a new virus was isolated from three American missionary nurses who worked in a town called Lassa and incurred a severe and previously undescribed hemorrhagic fever [65]. The identified so called Lassa virus (LASV), a member of the *Arenaviridae*, emerged to be a major public health problem in Western Africa during the following three years [66]. Here, *M. natalensis* was found to be the reservoir of this virus [66,67] and is still used as an animal model [68,69].

*Mastomys* are also highly susceptible to several helminths (nematodes, cestodes, trematodes) [70] and experiments with chemotherapeutics on *Schistosoma mansoni*-infected animals were performed already decades ago [71]. Filarial parasites like *Wuchereria bancrofti*, *Brugia malayi* and *Brugia timori* occupy the lymphatic system and cause the mosquito-borne disease lymphatic filariasis [72], characterized by long-term disability and severe immunopathology, e.g., elephantiasis [73]. Since available drugs only have low macrofilaricidal activity and have to be applied in long-term regimens, there is a need for new treatment and prevention strategies. In this context, *M. coucha* served as a permissive system for *B. malayi* infection to study immune responses [74,75,76,77], as well as a preclinical model for novel vaccines, including the use of recombinant filarial proteins [78,79,80,81] and DNA vaccination approaches [82].

*Mastomys* have also been reported to develop spontaneous adenocarcinomas and gastric carcinoids in high frequencies and were in turn used to study these stomach cancers [83,84,85,86,87,88]. In humans, gastritis and carcinoids can be triggered—amongst other reasons—by gram-negative bacteria of the genus *Helicobacter*. The main member *H. pylori* is prevalent in approximately 50% of the global human population (estimated 4.4 billion people) [89] and responsible for 75% of non-cardia gastric carcinomas worldwide [90]. Consequently, *H. pylori* was classified as a carcinogen for humans [91]. *Mastomys* have been used to study the effects of *H. pylori*-colonization on carcinoid formation [92]. Distributed throughout the gastrointestinal epithelium, so-called enterochromaffin (EC) cells are regarded as the predominant neuroendocrine cells of the bowel [93] and enterochromaffin-like (ECL) cells in the stomach regulate acid secretion [94]. As shown in a *Mastomys*-derived gastric enterochromaffin-like (ECL) cell neoplasia in vitro model, *H. pylori* lipopolysaccharides (LPS) have mitogenic effects on tumor ECL cells [95]. Notably, in 2005, during analyses of specimens from *Mastomys*, the novel *Helicobacter* species *H. mastomyrinus* has been isolated [96], which can cause severe inflammatory bowel disease in certain mouse strains [97].

Like mice and rats, *Mastomys* were shown to be susceptible to the Murine Sarcoma Virus-Harvey (MSV-H) and develop erythroblastic splenomegaly and large sarcomas upon infection [98]. In the past, *Mastomys* were also infected with autonomous parvoviruses (minute virus of mice prototype strain, MVMp and H-1) to evaluate their usability for parvovirus-based vectors. However, this was not found to be applicable, since both viruses were pathogenic for *Mastomys* while harmless for mice and rats, respectively [99].

Notably, in 2011 a novel polyomavirus (PyV) was identified in a wild African *Mastomys* that is phylogenetically closely related to the murine pneumotropic PyV [100] and was named *Mastomys* PyV (MasPyV). Like human polyomaviruses, e.g., BKV, JCV or MCPyV, causing hemorrhagic cystitis [101], progressive multifocal leukoencephalopathy [102] or Merkel-cell carcinoma [103], respectively, MasPyV persistently infect their host cells and do not cause pathological symptoms under immunocompetent conditions [100,104].

In the following sections, we will provide some historical and recent aspects about how the animal model *Mastomys coucha* contributes to the understanding of cutaneous papillomaviruses and their role in the development of non-melanoma skin cancer (NMSC).

## 5. *Mastomys coucha* as a Preclinical Model in Papillomavirus Research

### 5.1. History of Mastomys coucha in Papillomavirus Research

The use of *Mastomys coucha* in papillomavirus (PV) research is attractive, since it offers the possibility to study the function of a cutaneous PV in a natural infection model. The colony currently housed at the German Cancer Research Center (DKFZ) in Heidelberg (Germany) emerged from the chamois-colored, red-eyed GRA-Giessen strain that was held after 1966 at the Institute for Parasitology in Giessen (Germany). Their offspring were transferred to the DKFZ in 1969, because initially a high incidence of stomach cancer was noticed (Oettle, 1957). Whether these particular tumors were induced by *H. mastomyrinus* [96], by papillomaviruses [105] or by both is still an interesting question that remains to be clarified. Spontaneously appearing epithelial skin lesions in the GRA-Giessen strain were first described and initially classified as “so-called” keratoacanthomas [106,107]. Due to the endemic occurrence in the colony, a virus was suspected to be the etiological agent of these lesions.

Indeed, when transferring homogenized cell-free tumor tissue to scarified skin of different animals, according to the Koch postulates [108], new tumors indistinguishable from spontaneous tumors emerged [106]. Morphologically identical viral particles could be isolated from these lesions (papillomas and keratoacanthomas, KAs) [109] (Figure 4A,B), as well as from well-differentiated squamous cell carcinomas (SCCs) [110,111,112]. Serum obtained from rabbits immunized with purified virions could further prevent tumor formation in experimentally infected animals [112].

Viral particles can be found in large amounts in nuclei of tumor cells and keratinized masses of skin tumors [110], as well as upper parts of the *stratum granulosum* and *stratum corneum* [113] (Figure 4C). They were structurally characterized as a typical papillomavirus of 52 nm in size, referred to as *Mastomys natalensis* papillomavirus (MnPV) [112]. The sequence of the MnPV genome contains open reading frames for the early genes E6, E7, E1, E2 and E4, the late genes L1 and L2, as well as an untranslated URR (upstream regulatory region) (Figure 4D) [114]. The E5 gene, known from mucosal alpha-type human papillomaviruses (HPVs), is missing—a feature shared with cutaneous beta-type HPVs [115]. These, like HPV5 or HPV8, cause epithelial lesions or tumors especially in organ transplant recipients [116,117], but also in immunocompetent individuals [118].

Apparently, the so-called KAs observed in *Mastomys* (Figure 4B) differed from KAs known from humans regarding histology and time course. While, macroscopically, a strong keratinization and infiltration to surrounding tissue were similar, human KAs were described as emerging from hair follicles [119], while those in *Mastomys* seemed to arise from the surface and at best at the infundibulum of the hair follicle [109]. However, considering later data obtained by in-situ hybridization, MnPV as the etiological agent for papillomas, KAs and SCCs could be also detected in hair follicle cells, being apparently a reservoir for these viruses [105]. Up to date, a spontaneous regression described for human KAs was never reported for *Mastomys* where the tumor continues growing and infiltrating even down to the muscle tissue [110].

Human skin is an open ecosystem [120] and becomes colonized already shortly after birth by a wide range of cutaneous HPV types [121]. A homologous validity is true for *Mastomys coucha*, since MnPV DNA in normal skin, as well as seroconversion, can be detected at 4–5 weeks of age, probably, due to nursing between the mother and their offspring [122]. Due to a natural infection route within the colony and the occurrence of MnPV-positive skin lesions, *Mastomys* were found to be an ideal model system to investigate the contribution of MnPV to skin carcinogenesis. As shown in previous studies, persisting MnPV genomes in the skin become activated after administration of 7,12-dimethyl-benz(a)anthracene (DMBA) and/or 12-O-Tetradecanoylphorbol-13-acetate (TPA) resulting in the formation of both benign and malignant skin tumors [123,124]. The oncogenic potential of MnPV was also demonstrated in MnPV-E6 transgenic mice (E7 transgenic animals died *in utero* for unknown reasons), showing nearly a 100% formation of SCCs upon DMBA/TPA treatment compared to only 10% of their non-transgenic littermates [125]. Interestingly, while chemically induced SCCs in wildtype mice usually harbor specific DMBA-induced activating *Hras* mutations that favor cell growth upon TPA treatment [126], skin tumors obtained in MnPV-E6 transgenic mice consistently contained wild-type *Ras* at all three hot-spot positions [125]. Intriguingly, the same mutual exclusion of papillomavirus positivity and *Hras* mutations could be observed in SCCs from melanoma patients after treatment with BRAF inhibitors, such as Vemurafenib [127], indicating that cutaneous HPVs (similar to MnPV) may substitute or circumvent activating *Hras* mutations [128].

Chemical DMBA/TPA-induced skin carcinogenesis—although providing interesting insight into molecular mechanisms—do not really reflect physiological skin tumor development in humans [129]. More equivalent to tumor physiological promoting mechanisms was the observation that latent MnPV genomes in the skin can be activated by wounding or chronic mechanical irritation (repeated scratching of the skin with glass paper), finally leading to SCC formation [130]. Conversely, in the case of NMSC, the most important environmental risk factor, namely UV exposure, was still unnoticed at that time. However, transgenic mice [131,132] or animals naturally infected with genuine papillomaviruses [133,134] recently filled this gap, focusing on the functional interaction between PV infection and UV exposure. While a synergistic effect of UV and wound healing could be shown in transgenic HPV8-CER (complete early region) or HPV8-E6 mice [135], the effect of UV exposure on a natural PV infection was investigated later after the development of sensitive tools and methods needed to measure the course of a natural infection with minor amounts of viral DNA together with solid prospective serological data.

### 5.2. Recent Contributions of Mastomys coucha to PV Research as a Preclinical Model

While previous reports mainly focused on the final manifestation of skin tumors and the viral presence in those lesions, the subsequent studies in this research field were also considering the immunological consequences and the molecular mechanisms of the interplay between virus and host. Experimental techniques, such as PCR and qPCR, DNA/RNA sequencing and ELISAs, to monitor the immune response against individual viral proteins were not available during that time. Remarkably, the systematic analysis of body compartments by PCR of these animals showed that besides the skin, also other organs, e.g., stomach, lungs and liver were positive for MnPV DNA, while fetuses or newborns were negative [105]. Although viral DNA could also be detected in the blood, inner organs were not necessarily positive in the same animal. MnPV presence in the blood must be merely transient (temporary viremia), but apparently is the only way of viral spread to inner organs, because no other routes are plausible [105]. It turned out that the incidence of tumors strongly depends on the number of MnPV genomes in healthy skin, which enrich as the animal becomes older [105].

Analysis of viral gene expression in skin samples with productive infection identified a complex MnPV transcription pattern, revealing novel splicing isoforms that have not yet been described for other papillomaviruses [136], but can be further studied in the meanwhile in recently established *Mastomys*-derived keratinocyte and fibroblast cell lines [137]. Moreover, comparable to HPVs, two promoters for early and late transcripts and two different polyadenylation sites were identified within the MnPV genome. Estimation of expression levels of each transcript showed that L1 and E1^E4 mRNAs were the most abundant [136]. Consistent with episomal replication under permissive conditions, RNAseq did not reveal any viral-cellular fusion transcripts. The comprehensive transcription map provides a basis for understanding MnPV pathogenesis and allows the identification of transcripts expressed within tissue samples on a spatial level.

Moreover, improved detection methods also identified a new papillomavirus (McPV2) (Figure 5A), which is phylogenetically distant from MnPV and mainly found in anogenital lesions (Figure 5B,D), but also in other organs and mucosal tissues, such as the oral cavity or the tongue (Figure 5C,E) [138]. In that way, *Mastomys coucha* is the only model where the biology of both a cutaneous (MnPV) and mucosal papillomavirus (McPV2) can be investigated in the same animal. Here, as measured with glutathione S-transferase (GST)-capture ELISAs, the seroreactivity against the L1 capsid proteins of both viruses was found to be significantly increased in animals bearing the respective skin or anogenital lesions [139].

Papillomavirus serology of individual viral proteins additionally allows prospective studies in terms of the time course of infection and skin lesion development. Particularly antibodies against the early E2 proteins of both viruses already appear in one month old animals [122], while the immune response against the late L1 viral proteins is delayed, appearing after approximately 2.5 months, but indicative for productive viral infections [122]. Notably, in humans, seropositivity against cutaneous HPVs can also be detected already in early childhood [121], which at least in the case of certain types is associated with an increased risk for development of SCCs [140]. Seroconversion against MnPV L1 and E2 correlates with the manifestation of lesions and can be used as a marker of a current or a preceding infection. Hence, both serology and the appearance of lesions in the immunocompetent host-virus system *Mastomys coucha* offer the opportunity to test prophylactic and therapeutic strategies against infection with cutaneous papillomaviruses.

Nevertheless, although *Mastomys* coucha serves as a suitable natural model for different diseases including cancer, as in every model, it also has its limitations. Here, it is mainly the lack of standardized molecular and immunological tools, e.g., primer sequences for the measurement of gene expression levels or different immune responses. Established antibodies, for instance, although specific for mouse or/and rat, do not necessarily cross-react with *Mastomys* proteins and need to be carefully tested. Therefore, before investigating certain scientific questions with this animal—e.g., how infections are controlled, including the immunological surveillance of a commensal virus like MnPV and how this virus finally contributes to NMSC—it may be a challenge to establish needed tools. The future lab work is simplified, since the genome of *Mastomys coucha* was recently sequenced at the Institute for Human Genetics at the UCSF, San Francisco [141].

## 6. Vaccination and Tumor Prevention in *Mastomys coucha*

The success seen after immunization with virus-like particles (VLPs) against mucosal high-risk HPVs causing anogenital tumors also prompted the development of prophylactic vaccines against cutaneous PVs [142]. Especially for immunosuppressed organ transplant recipients, vaccination would be of great benefit, since up to estimated 40% of these patients develop NMSC within 10 years after transplantation and up to 80% after 20 years [143,144,145]. Their risk of developing such lesions is correlated with the HPV load in plugged eyebrow hair follicles [146]. The efficacy of such a vaccine on either ongoing or newly established MnPV infections was examined in the *Mastomys* model. For this purpose, an MnPV-free *Mastomys* colony was generated via hysterectomy to allow infections under defined experimental conditions. Using MnPV-VLPs, a strong and long-lasting immune response was established in virus-infected and virus-free animals even under systemic long-term immunosuppression. It comprised of high titers of neutralizing antibodies, as measured by pseudovirion-based neutralization assays [147]. In all cases, the formation of both benign and malignant skin lesions was completely prevented and further led to a significant reduction of MnPV DNA loads in normal skin [148]. This study provided the first evidence that a VLP-based vaccine can trigger an effective immune response in the skin irrespective of the immune and infection status at the time of vaccination. Thinking in translatability, this was an important proof-of-principle for the clinical development and application of vaccines against skin tumors caused by cutaneous HPV infection, especially in patients awaiting an organ transplant.

## 7. The Role of UV Exposure and Papillomavirus Infection in NMSC Development

Numerous seroepidemiological studies already support an association between infection with certain cutaneous HPV types and NMSC [140,149]. However, the sporadic absence of viral DNA within malignant lesions still raises skepticism whether HPVs (i) act via a “hit-and-run” mechanism, (ii) are just opportunistic bystanders or (iii) there are two independent events in the formation of virus-positive and -negative SCCs in the context of cumulative UV exposure as environmental risk factor [128].

In recent experiments, *Mastomys coucha* served as a model to decipher this fundamental question by studying the effect of UV exposure on MnPV-positive and -negative skin [134]. For this purpose, virus-free and naturally infected animals were chronically exposed to UVB light and the incidence of skin tumors was monitored. Notably, UVB irradiation (corresponding to UV doses of different geographical regions) only led to significant tumor formation in MnPV-infected skin. Two distinct tumor types were induced within a time frame and histopathology similar to humans (Figure 6A,B): Well-differentiated keratinizing SCCs (KSCC) that still support productive infections with high viral DNA loads and transcription and poorly differentiated non-keratinizing SCCs (nKSCC) (Figure 6C,D). The latter type may cause a microenvironment that counteracts viral propagation, which explains the low viral loads or even the lack of MnPV DNA and transcription [134]. Nevertheless, all tumor-bearing animals developed MnPV-specific antibodies, directly corroborating a preceding infection, a scenario also reported in SCC patients [150].

MnPV infection apparently leads to an attenuation of DNA repair and genomic stability and favors the accumulation of UV-induced mutations, including in *Trp53*. Especially two hot-spot positions—also well-known from human SCCs [151]—within the DNA-binding domain of p53 were found [134]. Since it is known that the loss of functional epidermal *Trp53* in mice can favor the expansion of poorly differentiated SCCs, a decisive role of mutated p53 on the phenotype of squamous cells can be anticipated [152]. In other words, functional loss of epidermal p53 in *Mastomys* tumors may favor dedifferentiation [153], which counteracts the permissive cycle and explains the differences in viral load between KSCCs and nKSCCs. Such a “hit-and-run” mechanism is a reasonable explanation for dispensable continuous viral presence during skin carcinogenesis as observed in patients with NMSC [128,134]. It provides the basis to investigate the temporal and spatial order of events underlying papillomavirus-driven skin carcinogenesis and for the development of preventive or curative strategies against NMSC.

## 8. Summary

As outlined above, particular scientific questions often require suitable animal systems. Although paradigmatic models like mice may be attractive and “well-accepted” within the scientific community, one should be open for nontraditional models that already exist in nature with great similarities to human diseases. The African rodent *Mastomys coucha* has a long history in various research fields, including parasite, bacteria and virus research. In the latter context, due to the high degree of resemblance with humans in terms of viral skin infections, *Mastomys coucha* is an easy-to-handle laboratory animal that represents a unique and powerful model to investigate basic molecular interactions between papillomaviruses and their natural host. Moreover, due to the presence of two papillomaviruses in the same animal, broad-protective vaccines [154] can be tested under different conditions.

## Figures and Tables

**Figure 1 viruses-11-00182-f001:**
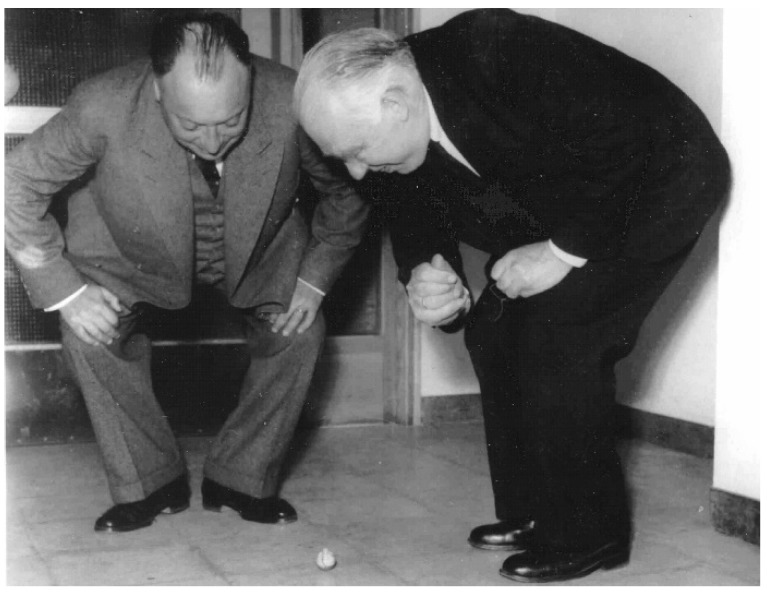
Wolfgang Pauli and Niels Bohr are watching a spinning top as a model for the spinning electron. Photograph by Erik Gustafson, courtesy of AIP Emilio Segré Visual Archives. Courtesy of the Margrethe Bohr collection, Kopenhagen.

**Figure 2 viruses-11-00182-f002:**
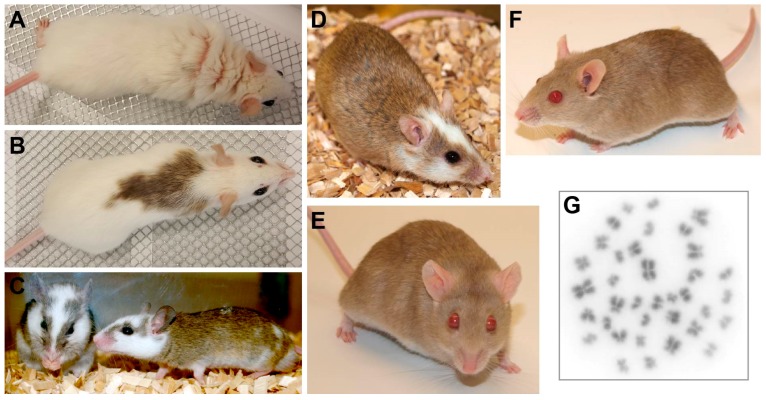
The multimammate mouse *Mastomys coucha*. (**A**–**D**) Black-eyed *Mastomys coucha* with different coat colors. (**E**,**F**) Chamois-colored red-eyed *Mastomys coucha* used at the DKFZ (derived from the GRA-Giessen strain). (**G**) Metaphase chromosome spread obtained from a *Mastomys coucha* splenocyte (2*n* = 36; 630× magnification).

**Figure 3 viruses-11-00182-f003:**
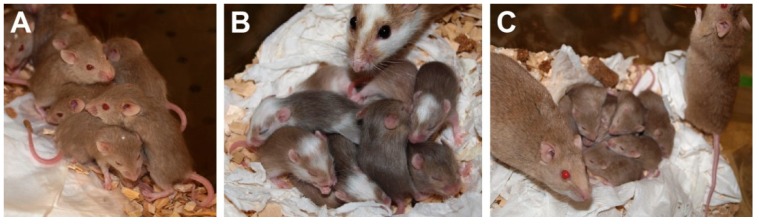
Young *Mastomys coucha*. (**A**) Young *Mastomys* usually huddle together. (**B**,**C**) Parental *Mastomys*, especially the mother, have a strong protective instinct and always stay close to their offspring.

**Figure 4 viruses-11-00182-f004:**
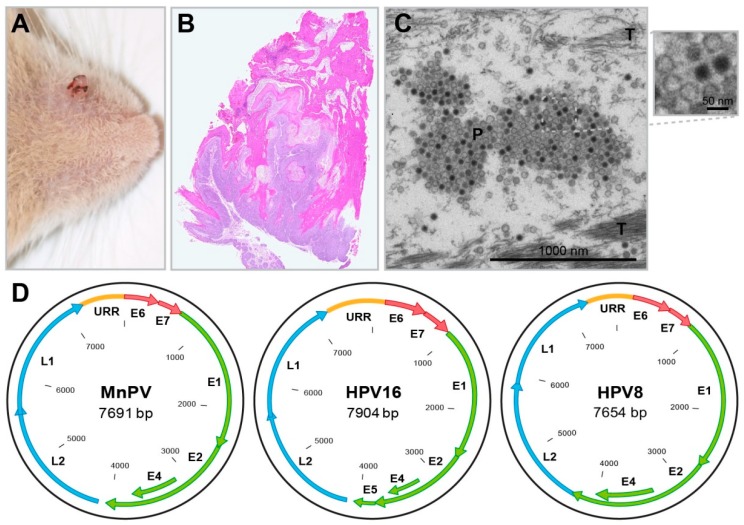
The *Mastomys natalensis* papillomavirus. (**A**) A spontaneous skin tumor (papilloma) near the nose. (**B**) HE staining of a spontaneous skin tumor (“so-called” keratoacanthoma) shows typical endo-exophytic growth and strong keratinization. (**C**) EM micrograph of MnPV particles (P) in the most upper layer of a spontaneous MnPV-induced skin lesion. While larger host cell compartments are already degraded during terminal differentiation and desquamation, tonofibrils (T) are still visible. (**D**) Schematic representation of the genomes of HPV16 (an alpha-type), HPV8 (a beta-type) and MnPV (an iota-type). While the genomes of all PV types harbor an upstream regulatory region (URR) and code for early genes (E1, E2, E4, E6, E7) and late genes (L1, L2), PVs from the genus beta and iota do not contain E5 open reading frames.

**Figure 5 viruses-11-00182-f005:**
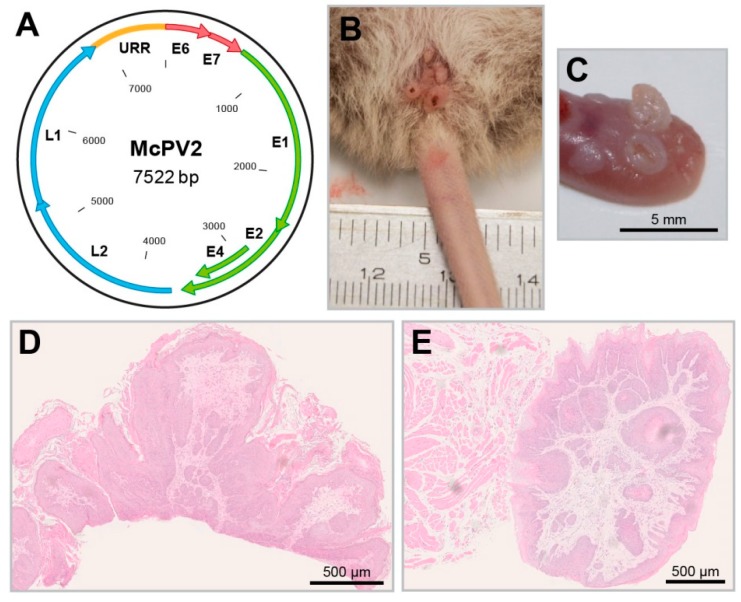
The *Mastomys coucha* papillomavirus 2. (**A**) Schematic representation of the McPV2 genome. (**B**) Condylomas in the anogenital region induced by McPV2. (**C**) Tongue papillomas induced by McPV2 (these are frequently positive for MnPV as well). (**D**) HE
staining of a condyloma. (**E**) HE staining of a tongue papilloma.

**Figure 6 viruses-11-00182-f006:**
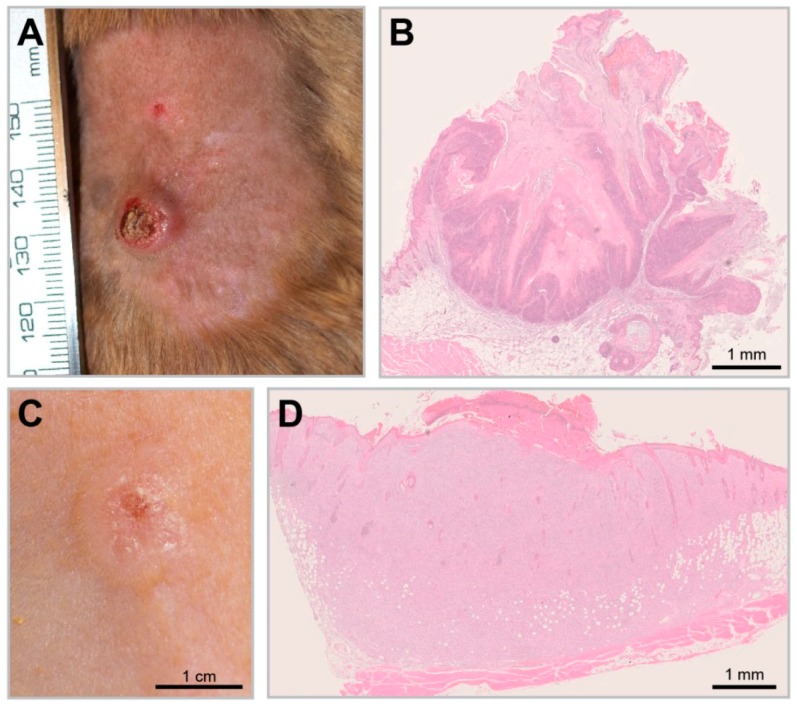
UV-induced SCCs in the *Mastomys* model. (**A**) A UV-induced KSCC. (**B**) KSCCs are characterized by the growth of well-differentiated squamous cells and show strong keratinization comparable to spontaneous MnPV-induced tumors (HE staining). (**C**) A UV-induced nKSCC. (**D**) HE staining of a poorly differentiated nKSCC.

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
