# Peer review of "Mastomys Species as Model Systems for Infectious Diseases"

_viruses, 2019, doi:10.3390/v11020182_

Round 1
Reviewer 1 Report
Review:Mastomys species as model systems for infectious diseases.
Daniel Hasche and Frank Rösl
In this review, the authors take a look at the use of Mastomys coucha as a preclinical animal model. They explore the past and current uses of this animal model along with its strengths and particular interest for papilloma virus (PV) research.
Overall, this was a very good review. It was informative and interesting and was an appropriate length. I believe the subject matter is relevant to the journal and would interest many readers. I only have a few small comments regarding the manuscript. I noticed a few grammar/sentence structure issues (listed below). I also have a few comments/questions which might help improve the manuscript.
General questions/comments:
1) On lines 14-15, the authors state that it is mandatory to broaden our view by also using non-traditional animal models. This is not further explained. Would they like to elaborate on the meaning of this sentence?
2) The authors could state the other alternate names for the mastomys such as African soft- furred rat.
3) On lines 64-65, The authors discuss other animal models used in research such as zebrafish, rats, dogs, pigs, goats and monkeys. Please add references for these.
4) Could the authors please elaborate more on the different strains of mastomys and why they are suitable or unsuitable for research if known? How does one select the most appropriate strain for their research question?
5) In Figure 2 G the chromosomes are 2n=36 how does that compare to other species (the mouse has 40) ? Has the genome of mastomys been entirely sequenced?
6) Lines 184-186, I feel it is unnecessary to introduce the next section of the review.
7) Lines 369-374, Summary. It would be great to have a broader conclusion and summary of the review. Reiterate the uses and strengths of mastomys and summarise the whole body of work not just the PV part of it. Also, perhaps leaving the reader with a take home message or new outlook.
Grammar (highlighted words have been corrected):
1) Lines 28-29, Models are only needed as long as we do not yet have full knowledge of what they stand for. This sentence is awkward and confusing consider rewriting.
2) Lines 34-35, (…) that some observations in research cannot be represented in any other way than in the form of models.
3) Line 41 A model also stands as asubstitute
4) Lines 80-85, Very long sentence, consider breaking it up into 2 sentences or shortening
5) Lines 87-89 (…) in the scientific community it has been known for several decades in various research fields and has recently attracted great attention in functional studies of cutaneous papillomaviruses and non-melanoma skin cancer formation.
6) Line 144, (…) a member of the Arenaviridae, emerged to bea major public health problem.
7) Lines 157-170, I am unsure why these are divided into 2 separate paragraphs.
8) Please verify the use of mastomys sometimes it is used as third person singular and sometimes plural and switches back. See lines 172-183. Please verify the right person is being used.
9) Lines 184-186, I already suggested this be deleted but if it is not please change (…) historical and recent aspects about how the animal model Mastomys coucha contributes …
10) Lines 212 HPV the abbreviation was not introduced. Please write human papillomavirus out the first time.
11) Lines 234, Do the authors mean Human skin is an open ecosystem and becomes colonized shortly after birth by …
12) Line 301, please add a comma after in humans. Notably, in humans, seropositivity against …
Author Response
Manuscript ID.: viruses-447844
“Mastomys species as model systems for infectious diseases”
Dear Editor,
Thank you for evaluating our manuscript and for giving us the opportunity to submit a revised version to “Viruses”.
In the following, we thoroughly address all concerns of the referees and describe the modifications we have included in the revised version.
Reviewer #1:
“Overall, this was a very good review. It was informative and interesting and was an appropriate length. I believe the subject matter is relevant to the journal and would interest many readers. I only have a few small comments regarding the manuscript. I noticed a few grammar/sentence structure issues (listed below). I also have a few comments/questions which might help improve the manuscript.”
We estimate the comment of this reviewer and appreciate his/her help to improve the manuscript
General questions/comments:
1) “On lines 14-15, the authors state that it is mandatory to broaden our view by also using non-traditional animal models. This is not further explained. Would they like to elaborate on the meaning of this sentence?”
We further explained the meaning of this statement in the main text
2) “The authors could state the other alternate names for the mastomys such as African soft- furred rat.”
We have included it
3) “On lines 64-65, The authors discuss other animal models used in research such as zebrafish, rats, dogs, pigs, goats and monkeys. Please add references for these.”
We have included all references
4) “Could the authors please elaborate more on the different strains of mastomys and why they are suitable or unsuitable for research if known? How does one select the most appropriate strain for their research question?”
This is a justified question that is hard to answer. It is known that certain genomic susceptibility loci play a role, but this has not yet been addressed for the different Mastomys species. Nevertheless, please note that a similar situation is also reported for mice, since some mouse strains are more suitable for expressing a transgenic phenotype like others. Please see: Taketo et al., 1991; Proc Natl Acad Sci U S A. 88(6): 2065–2069.
5) “In Figure 2 G the chromosomes are 2n=36 how does that compare to other species (the mouse has 40) ? Has the genome of mastomys been entirely sequenced?”
Yes, mouse and Mastomys have different chromosomes and indeed, the Mastomys coucha genome is entirely sequenced in the meantime. We have included a link in the revised manuscript.
6) “Lines 184-186, I feel it is unnecessary to introduce the next section of the review.”
We would like to leave it as it is, if this referee doesn´t mind.
7) “Lines 369-374, Summary. It would be great to have a broader conclusion and summary of the review. Reiterate the uses and strengths of mastomys and summarise the whole body of work not just the PV part of it. Also, perhaps leaving the reader with a take home message or new outlook.”
We have considered this point and expand the outlook in the revised manuscript.
Grammar (highlighted words have been corrected):
1) “Lines 28-29, Models are only needed as long as we do not yet have full knowledge of what they stand for. This sentence is awkward and confusing consider rewriting.”
We explained the meaning of this sentence to avoid confusion
2) “Lines 34-35, (…) that some observations in research cannot be represented in any other way than in the form of models.”
corrected
3) “Line 41 A model also stands as a substitute”
corrected
4) “Lines 80-85, Very long sentence, consider breaking it up into 2 sentences or shortening”
corrected. Two sentences and a better explanation is provided
5) “Lines 87-89 (…) in the scientific community it has been known for several decades in various research fields and has recently attracted great attention in functional studies of cutaneous papillomaviruses and non-melanoma skin cancer formation.”
corrected
6) “Line 144, (…) a member of the Arenaviridae, emerged to be a major public health problem.”
corrected
7) “Lines 157-170, I am unsure why these are divided into 2 separate paragraphs.”
corrected
8) “Please verify the use of mastomys sometimes it is used as third person singular and sometimes plural and switches back. See lines 172-183. Please verify the right person is being used.”
Mastomys: it´s a singular-only noun, and plural is identical, like “fish” or “aircraft”
9) “Lines 184-186, I already suggested this be deleted but if it is not please change (…) historical and recent aspects about how the animal model Mastomys coucha contributes …”
corrected
10) “Lines 212 HPV the abbreviation was not introduced. Please write human papillomavirus out the first time.”
corrected
11) “Lines 234, Do the authors mean Human skin is an open ecosystem and becomes colonized shortly after birth by …”
Yes, we specified this statement
12) “Line 301, please add a comma after in humans. Notably, in humans, seropositivity against …”
corrected
In conclusion, we like to thank the reviewer for the critical and helpful comments. We trust that we addressed all concerns and changed our manuscript accordingly to the suggestions.
Sincerely yours
Frank Rösl
Reviewer 2 Report
This manuscript titled "Mastomys species as model systems for infectious diseases" by Daniel Hasche and Frank Rösl comprehensively and timely reviewed the current state of Mastomy models for infectious diseases. The review includes general comments on animal models of human diseases, physiology of Mastomys and their phylogenetic relationship to other rodents, historic perspective of the models and current utilities of these models for studying human infectious diseases. In short, I think this is a very good review. I suggest the authors to discuss potential limitations of this models, as all the animal models have some limitations as authors also stated.
Author Response
Manuscript ID.: viruses-447844
“Mastomys species as model systems for infectious diseases”
Dear Editor,
Thank you for evaluating our manuscript and for giving us the opportunity to submit a revised version to “Viruses”.
In the following, we thoroughly address all concerns of the referees and describe the modifications we have included in the revised version.
Referee #2
Comments and Suggestions for Authors
“This manuscript titled "Mastomys species as model systems for infectious diseases" by Daniel Hasche and Frank Rösl comprehensively and timely reviewed the current state of Mastomy models for infectious diseases. The review includes general comments on animal models of human diseases, physiology of Mastomys and their phylogenetic relationship to other rodents, historic perspective of the models and current utilities of these models for studying human infectious diseases. In short, I think this is a very good review. I suggest the authors to discuss potential limitations of this models, as all the animal models have some limitations as authors also stated.”
We thank this referee for his/her estimation of our review. We now have included also the limitations of the Mastomys model
In conclusion, we like to thank the reviewer for the critical and helpful comments. We trust that we addressed all concerns and changed our manuscript accordingly to the suggestions.
Sincerely yours
Frank Rösl